# Analysis of Acoustic Emission Signals Recorded during Freeze-Thaw Cycling of Concrete

**DOI:** 10.3390/ma14051230

**Published:** 2021-03-05

**Authors:** Libor Topolář, Dalibor Kocáb, Luboš Pazdera, Tomáš Vymazal

**Affiliations:** Faculty of Civil Engineering, Brno University of Technology, Veveří 95, 602 00 Brno, Czech Republic; Dalibor.Kocab@vutbr.cz (D.K.); pazdera.l@fce.vutbr.cz (L.P.); Tomas.Vymazal@vutbr.cz (T.V.)

**Keywords:** acoustic emission method, freeze-thaw cycles, concrete, signal analysis, short-time Fourier transform, fast Fourier transform

## Abstract

This manuscript deals with a complex analysis of acoustic emission signals that were recorded during freeze-thaw cycles in test specimens produced from air-entrained concrete. An assessment of the resistance of concrete to the effects of freezing and thawing was conducted on the basis of a signal analysis. Since the experiment simulated testing of concrete in a structure, a concrete block with the height of 2.4 m and width of 1.8 m was produced to represent a real structure. When the age of the concrete was two months, samples were obtained from the block by core drilling and were subsequently used to produce test specimens. Testing of freeze-thaw resistance of concrete employed both destructive and non-destructive methods including the measurement of acoustic emission, which took place directly during the freeze-thaw cycles. The recorded acoustic emission signals were then meticulously analysed. The aim of the conducted experiments was to verify whether measurement using the acoustic emission method during Freeze-thaw (F-T) cycles are more sensitive to the degree of damage of concrete than the more commonly employed construction testing methods. The results clearly demonstrate that the acoustic emission method can reveal changes (e.g., minor cracks) in the internal structure of concrete, unlike other commonly used methods. The analysis of the acoustic emission signals using a fast Fourier transform revealed a significant shift of the dominant frequency towards lower values when the concrete was subjected to freeze-thaw cycling.

## 1. Introduction

Concrete has been used in construction practically since the discovery of cement. During their service life, concrete elements are exposed to various degradation factors, such as mechanical and chemical influences, rapid temperature changes, etc. The knowledge of the influence of temperature changes on the quality of concrete during its service life is crucial both from the scientific point of view and in particular from the point of view of practical application in construction. In addition to the degrading impacts of very high temperatures, the alternation of positive and negative temperatures is one of the most destructive operating factors for a number of concrete products [1,2]. Freeze-thaw (F-T) cycles may adversely, as well as very quickly affect the durability of concrete structures. From a research point of view, it is appropriate to monitor the behaviour of concrete already during the actual exposure to freeze-thaw cycles [3,4,5].

In the case of concrete, which is in contact with water (examples of such structures include railway sleepers, road panels, water tanks, etc.), capillarity causes water to enter the pore structure of the concrete. At negative ambient temperatures, water begins to turn into crystalline ice and then to ice, which has a larger volume than water by about 9%. This presence of water inside the concrete structure exerts pressures that can significantly and irreversibly damage the concrete [6].

The assessment of the resistance of concrete to freeze-thaw cycles is typically based only on monitoring of certain mechanical properties, such as strengths, moduli of elasticity, etc. These properties are obtained after a certain number of F-T cycles using destructive or non-destructive methods. Subsequently, the observed changes in the properties (often a decrease) are evaluated with regard to the values obtained before freezing [7]. In the case of destructive tests, the sample is therefore not monitored continuously, but the behaviour of concrete is assessed on the basis of the results of individual test specimens. Despite the fact that the test specimens comprise one set of identical specimens produced from the same concrete, this can lead to inaccurate conclusions. This is due to the fact that the estimation of concrete behaviour is based on a statistical interpretation of the results of similar, but not exactly identical, specimens. When using non-destructive testing methods [8,9,10,11], monitoring of the condition of a concrete sample is conducted on the same specimens—the change of the monitored property is observed on individual specimens during the test at discrete time intervals (always after a certain number of completed F-T cycles). This process exhibits a lower degree of error than destructive testing methods. However, it is ideal to monitor the test specimens continuously even during freeze-thaw cycles, i.e., throughout the entire test. The acoustic emission method allows such an approach. The analysis of the recorded acoustic emission signals can be successfully used for a more detailed evaluation of the behaviour of the tested materials [12,13].

Acoustic emission (AE) is a real-time non-destructive testing method which can be employed to monitor the formation of cracks in concrete [14]. AE signals correspond to sound waves that emerge during the formation of cracks in a material. When a crack forms in concrete, energy is released and part of this energy is scattered in the form of an acoustic wave [15,16]. Conversely, acoustic waves and the corresponding energy are released when material damage occurs due to freeze-thaw cycles [4,17,18]. The AE signal parameters (e.g., number of counts, amplitude, frequency, etc.) can provide an effective tool to determine the degree of material damage during F-T cycles [19]. In general, the power of AE signals and their parameters depend on the amount of the released energy, the source, the distance and orientation of the source in relation to the location of the AE sensors [20]. Testing using AE relies on AE sensors continuously recording AE signals that are generated by the formation of damage in a material during its loading (e.g., F-T cycles). In cement-based composite materials, the source of AE activities can be found either in the cement binder or in the interfacial transition zone (ITZ) [20].

The aim of this manuscript is to compare the behaviour of test specimens of different sizes (produced from core samples drilled from a concrete block) during freezing and thawing. The interpretation of the frost resistance of concrete will be performed using the outputs of the acoustic emission method, resp. by analysing AE signals generated during 100 F-T cycles. For comparison, results of traditional non-destructive and destructive tests performed after every twenty-fifth F-T cycle will also be presented.

Another main goal of the described experiment was to determine a way to record the first changes to the quality of concrete, or the first signs of damage to the internal structure of concrete during exposure to F-T cycles. For this reason, air-entrained concrete was selected for the experiment since it was expected to exhibit a lesser degree of damage when exposed to F-T cycles and, above all, very gradual progression of the damage.

## 2. Experiment Description and Setup

The composition of the employed concrete is given in Table 1. The water/cement coefficient of the concrete was 0.46. The basic properties of fresh concrete were determined: density according to [21] was 2290 kg/m^3^, flow according to [22] was 460 mm, slump according to [23] was 180 mm, air content according to [24] was 5.0% and the temperature of the fresh concrete was 28 °C.

A 2.4 m high concrete block with floor plan dimensions of 1.8 m × 0.45 m was produced from the air-entrained concrete in an exterior in an open space. Concreting was conducted vertically into the wall formwork and the concrete was compacted using an immersion vibrator. After concreting, the block was covered with a damp cloth and then a PE foil. During the first two days after concreting, the cloth under the PE foil was regularly moistened. The concrete block was left in the formwork for one week, the formwork was then removed, and the concrete of the block was not treated any further.

Approximately two months after concreting, core samples for the production of the test specimens were drilled from the concrete block and had a diameter of 100 mm or 150 mm and the same length of 450 mm (block width). Cylinder and prism test specimens were then cut from the core samples using a diamond circular saw with constant water cooling. The described experiment included 4 sets of test specimens—cylinders with a diameter and length of 100 mm (marked CS100), cylinders with a diameter and length of 150 mm (marked CS150), cylinders with a diameter of 100 mm and length of 200 mm (marked C200) and prisms with dimensions 95 mm × 95 mm × 380 mm (marked P95, cut from the core sample with a diameter of 150 mm). Each set comprised 9 test specimens, which were divided into three groups of three. The first group were reference specimens, which were not exposed to freezing and thawing. The second group of specimens was subjected to 50 F-T cycles, and the last third group of specimens was subjected to 100 F-T cycles. Different properties and their development in relation to the number of conducted F-T cycles were monitored on the individual sets of the test specimens. Tensile splitting strength, according to [25], was determined on the CS100 and CS150 sets and these specimens were also subjected to the measurement using the AE method during F-T cycles. In the case of the C200 and P95 sets, the relative dynamic modulus of elasticity (RDM) was determined using the ultrasonic pulse velocity method and the resonance method, according to [26,27]. The P95 set was also subjected to the determination of flexural strength according to [28]. At least 3 test specimens were used for all destructive tests (even more in the case of non-destructive tests). Only continuous measurement of acoustic emission involved two specimens for each specimen size due to a limited number of AE sensors. Despite that, the results from both sensors on one specimen size did not differ statistically significantly.

The frost resistance test of concrete was conducted according to the standard [29]. The selected procedure is less time demanding than, for example, the procedure according to [26], the temperature range is larger than, for example, in the procedure according to [27] (which theoretically accelerates the concrete degradation process) and the freezing and thawing process can be very easily repeated since the times of the individual cycle parts are strictly defined. Equipment KD 20 (manufactured by EKOFROST s.r.o., Olomouc, Czech Republic, see [30]) was used for the test since it allows setting of the required freezing and thawing intervals and the test is performed automatically. One F-T cycle consists of freezing in air at −15 °C to −20 °C (the negative temperature we selected was always −18 °C) and thawing in a water bath at +20 °C. Freezing to the desired temperature takes 0.5 h and the negative temperature is then maintained for 3.5 h. Heating in a water bath takes 2 h. The total time of one F-T cycle is therefore 6 h + approximately 15 min (filling and draining water into the vessel of the KD 20 equipment where the test specimens are placed). The test was interrupted after every 25 F-T cycles, which lasts approximately one week, and respective test specimens were subjected to measurement of RDM or tensile splitting strength and flexural strength (after 50 and 100 FT cycles), the measured data were saved, and recording of the measurement using acoustic emission was re-initiated.

The measurement using the AE method was conducted continuously during cycling. One AE sensor was glued to each upper face of two CS100 test specimens and two CS150 test specimens, see Figure 1. The glue bond of the sensors was checked after every 25 F-T cycles.

The acoustic emission activity was generated by material damage during F-T cycles. Monitoring the AE activity was done by a multi-channel unit DAKEL XEDO (ZD Rpety-Dakel, Rpety, Czech Republic) [31] with the following input parameters: threshold value for counts was 200 mV, for individual AE hits then 400 mV, sampling of AE hits was set to 4 MHz, frequency range from 10–2000 kHz. The utilised sensors are hermetically sealed and have an IP68 degree of protection with increased frost resistance. The sensors are equipped with an integrated preamplifier and the total gain was 65 dB. The use of these extremely wideband sensors is intentional, in particular to filter out unwanted signals during postprocessing. The sensors do not have a model number because they were custom made in a limited number by the company DAKEL. The test also employed two monitoring sensors, which were placed on materials whose structure is not influenced by F-T cycles. These monitoring sensors were used to filter out false signals from the environment during post-processing. This approach provided pure signals from the individual test specimens. The AE sensors were attached to the specimens with an ethyl-based adhesive—it is a rubber-filled, resilient product with increased flexibility and peel resistance, with moisture resistance and a temperature range for use from −40 °C to +100 °C. The AE sensors, including fine-tuning of their mounting, were thoroughly tested in 2013, well before the first freezing experiments were conducted, see [4]. The first experiments also employed mechanical mounting, which we subsequently abandoned as it was not necessary. The AE sensors are tested and inspected every year and their characteristics remain unchanged. One problem occurred during the experiment and related to the computer’s internal memory. Although it seemed that everything was progressing correctly and the record was being saved, the actual situation was different. Unfortunately, it was not possible to retrieve data from the interval between the 25th and 50th F-T cycle—the data were unusable.

As has already been mentioned, after every 25th F-T cycles, RDM was determined according to the relation:(1)$$RDM=\frac{X_{N}^{2}}{X_{0}^{2}}\cdot 100\%,$$
where *RDM* is the relative modulus of elasticity of the concrete in %, *X_N_* is the respective dynamic quantity after N performed F-T cycles and *X*_0_ is the same quantity on the same test specimen before the start of the frost resistance test [26]. The dynamic quantity in this case is either the ultrasonic pulse velocity (UPV) in km/s, or the natural frequency of longitudinal oscillation f_L_ in kHz. UPV was measured using a Pundit PL-200 instrument manufactured by Proceq SA (Proceq AG, Schwerzenbach, Switzerland) [32] with 150 kHz probes. Each test specimen was measured in three longitudinal lines and UPV was determined as the average of these three measurements. The determination of f_L_ was done using a Handyscope HS4 oscilloscope manufactured by TiePie engineering (Sneek, Netherlands) [33].

Strength was determined on 3 test specimens from each set—either tensile splitting strength (CS100 and CS150) or flexural strength (P95) before the start of freezing and thawing, after the 50th and then after the 100th F-T cycle. The results were used to calculate the relative strength (RS) in a manner analogous to RDM,
(2)$$RS=\frac{f_{N}}{f_{0}}\cdot 100\%$$
where *RS* is the relative strength in %, *f_N_* is the average strength (tensile splitting or flexural) after N performed F-T cycles, and *f*_0_ is the same average strength found on 3 specimens not exposed to F-T cycles. The relative flexural strength *RS*(*F*) is the main criterion for the evaluation of frost resistance of concrete to F-T cycles in the standard [29]. The strength tests were conducted in a DELTA 6-300 testing machine manufactured by FORM+TEST Seidner and Co. GmbH (Riedlingen, Germany) [34].

AE signals are evaluated, for example, on the basis of counts, amplitude height, frequency, etc. The least complicated approach is to add up the counts over a threshold level. This threshold can be set, which allows control of the minimum threshold, and if exceeded, i.e., one count, this forms one pulse, which is counted by the counter. One AE hit can create several counts and their number depends on the set threshold level [35]. When counting an AE hit, it is necessary to rectify and filter the high frequency pulse of the hit. The number of counts is a so-called cumulative parameter, from which cumulative curves are obtained. The frequency and amplitude band of acoustic emission is wide, from units in Hz to high ultrasonic frequencies in MHz [36]. The pulse shape and the decrease in amplitude depend on the geometry of the test specimen and on its material properties, or on the degree of material damage. To determine the cause of an AE hit, it is necessary to perform a frequency analysis of the spectrum of the recorded AE signal. AE sensors are designed to receive surface waves, which are then converted to electrical signals. These signals are amplified, filtered and saved. The measuring process of the AE system begins at the moment when the value of the amplified and filtered analogue signal exceeds the set threshold level.

## 3. Results

The results of tensile splitting and flexural strengths are shown in Table 2 and Table 3 then presents the relative expression of not only the strengths (RS) but also the moduli of elasticity (RDM) in relation to the number of performed F-T cycles. Mechanical properties that are obtained in a destructive way (i.e., on different test specimens) only estimate concrete behaviour—they certainly do not reflect the behaviour of individual test specimens during freezing and thawing. This may lead to inaccurate test results. Table 3 indicates that the F-T cycles had practically no impact on the tensile splitting strength of the concrete. Flexural strength was influenced by freezing and thawing to a greater degree since after 100 F-T cycles, the strength dropped to almost 85% of its original value. However, this significant decrease, when compared to the other monitored properties of the concrete, is partly caused by one specimen that achieved much lower strength than the other two specimens in the set. This may, thus, constitute the already mentioned inaccuracy in the evaluation of concrete.

On the other hand, relative dynamic moduli of elasticity, as an example of non-destructive testing, characterize the behaviour of test specimens during an entire frost resistance test of concrete. However, they do so only at discrete time points (at the moment of measurement after N F-T cycles), not continuously. With regard to the dynamic modulus of elasticity, the results in Table 3 show that the F-T cycles did not practically affect the concrete at all. The maximum recorded decrease amounted to less than 3 percentage points.

In contrast to the abovementioned tests, the AE method allows monitoring of the given state of each test specimen throughout the entire test. The AE method therefore indirectly describes the behaviour of test specimens during F-T cycles. To evaluate the acoustic emission records, the number of counts (over the set threshold level) in its cumulative form (AE_cum_) during F-T cycles was initially selected. The course of the cumulative number of counts over time shows AE activity, with higher AE activity corresponding to higher damage/failure, as is shown for example in [37]. The graph in Figure 2 presents a very similar character of AE_cum_ courses during the first 25 F-T cycles for both sizes of the test specimens. The cumulative curves for individual specimens do not differ significantly even in the later course of F-T cycles. This was one of the reasons why the experimentally obtained data of cumulative counts were interpolated with straight lines. The slope of the lines for both sizes of the specimens did not differ much during the first 25 F-T (Table 4). A change in the slopes of the interpolated lines occurs only during 50–75 F-T cycles, when the increase for larger specimens CS150 is significantly higher. Bearing in mind that the mechanical properties have not been affected (Table 3), this indicates a formation of small failures in large numbers. Nevertheless, in the last stages of the freezing cycles (from 75 to 100 F-T cycles), the smaller specimens CS100 exhibit a higher growth of AE activity, as can be seen in Table 4.

In the case of CS100 specimens, it can be further concluded from Figure 2 that the character of the slope from the third series of F-T cycles is maintained even at the beginning of the fourth series of F-T cycles, i.e., after the 75th F-T cycle. Approximately at the 85th F-T cycle, the slope begins to break to a higher value than shown in Table 4 (that is why that moment shows the lowest value of the coefficient of determination).

It can be stated that the most commonly used AE parameter in general, which AE counts undoubtedly is, indicated in this case only the breaking points of the ongoing damage to the material. Therefore, it was appropriate to conduct a more detailed analysis of the individual AE hits. Focus was directed at the height of the amplitude, the change of the position of the dominant frequency, and the attenuation of the spectral density of the individual AE hits.

The amplitude of the recorded AE hits indicates the degree of the emerging material damage. More significant damage to the material structure generates a higher signal amplitude, as shown for example in [38]. The height of the amplitude (see Figure 3) now shows more significant differences between the individual sizes of the tested specimens. While the amplitudes remain the same, within the measurement error, during the first 25 F-T cycles, a higher amplitude was recorded for the CS100 specimens in the subsequent cycles. This trend may indicate a greater degree of damage to these smaller specimens. This in fact corresponds to the changes of the slope in Table 4, respectively in the graph in Figure 2.

Another parameter that can be analysed from AE hits is the dominant frequency, which is obtained using the fast Fourier transform (FFT) from the time spectrum, which is also utilised in [39]. The graph in Figure 4 demonstrates how the dominant frequency for each recorded AE hit shifts towards low values during freezing and thawing. This shift is caused by the mechanical wave passing through an increasingly more damaged environment [40,41]. It can be observed that during the first 25 F-T cycles, the dominant frequencies reach values around 200 kHz and as the number of F-T cycles increases, the dominant frequency decreases. In the case of CS100 specimens, these average values are gradually 194, 142 and 61 kHz, and in the case of CS150 specimens then 199, 162 and 72 kHz.

The following series of evaluations, include typical AE hits recorded in individual stages of the F-T cycles for the individual sizes of the test specimens. The time-frequency spectrum of the power spectral density [42] (Figure 5c, Figure 6c, Figure 7c, Figure 8c, Figure 9c and Figure 10c) was calculated using a Short Time Fourier Transform with a Kaiser window. On the horizontal axis, the graph shows the time shift of the frequency, which is projected on the vertical axis. The size of the power spectral density is determined by the colour tones. The decibel scale is basically a linearization of the logarithm, i.e., [43],
(3)$$x_{dB}=20\cdot log\left(\frac{x_{V}}{x_{k}}\right),$$
where *x_dB_* is the value in decibels, *x_V_* is the value (in this case) in Volts and *x_k_* is the reference value also in Volts.

The Short Time Fourier Transform (STFT) spectrum is defined as [43],
(4)$$S\left(\tau ,f\right)=\overset{\infty }{\underset{-\infty }{\int }}x\left(t\right)w\left(t-\tau \right)e^{-i2\pi ft}dt$$
where *w*(*τ*) is the window function (Kaiser in this case), *x*(*t*) is the observed function, *S*(*τ*,*f*) is the resulting complex spectrum, *f* is the frequency and *τ* is the time shift (location).

Evaluation uses the absolute spectrum value [43],
(5)$$\left|S\left(\tau ,f\right)\right|=S\left(\tau ,f\right)\cdot S{\left(\tau ,f\right)}^{*}$$
where *S*(*τ*,*f*)* is the complex combined function.

Selected typical AE signals recorded in CS100 specimens are presented in graphs in Figure 5, Figure 6 and Figure 7 (all presented graphs are provided only for illustration and better understanding for the reader).

The series of graphs in Figure 5, relating to CS100 specimens during the first 25 F-T cycles, shows that the maximum spectrum values at the beginning of the signal (frequency band 120–230 kHz) are relatively quickly attenuated. With regard to the most significant frequency of 201 kHz, attenuation of spectral density reaches 62 dB/(Hz∙ms). The longest occurring frequency in the signalis 70 kHz, for approximately 1 ms.

The series of graphs in Figure 6, which apply to CS100 specimens from the 50th to the 75th F-T cycle, includes significant values of the frequency spectrum in the frequency range 50–350 kHz, however, the range 50–220 kHz is the most important region. At these frequencies, the signal is attenuated very slowly. Attenuation of spectral density reaches 11 dB/(Hz∙ms) at the most significant frequency of 184 kHz. The longest sounding signals were found in the frequency range 170–210 kHz (for approximately 3 ms) and then in the frequency region around 120 kHz (for approximately 2.5 ms).

The time-frequency spectrum, see the series of graphs in Figure 7, reveals two significant frequency bands 50–100 kHz (attenuated after more than 3 ms) and 170–220 kHz (attenuated slightly below 3 ms). Attenuation of spectral density of the most significant frequency of 66 kHz was 4 dB/(Hz∙ms). These data apply to CS100 specimens during the last 25 F-T cycles.

Selected typical AE signals recorded from larger specimens CS150 are shown in the graphs in Figure 8, Figure 9 and Figure 10 (all presented graphs are provided only for illustration and better understanding for the reader).

The series of graphs in Figure 8, describing CS100 specimens during the first 25 F-T cycles, indicates that the significant frequency band is 50–300 kHz, of which the region between 160–250 kHz is extreme with regard to amplitudes. It is evident that the attenuation at higher frequencies is greater than at lower frequencies. Attenuation of spectral density at the most significant frequency of 185 kHz is 43 dB/(Hz∙ms). The signal length is relatively short, approximately up to 1.5 ms.

The significant frequency spectrum, see the series of graphs in Figure 9, is in a very narrow band 140–230 kHz but exhibits relatively long duration—more than 3 ms. At the most significant frequency of 171 kHz, attenuation of spectral density is 6 dB/(Hz∙ms), however, the spectrum shows apparent rapid attenuation—approximately 1 ms. These data apply to CS150 specimens between the 50th and 75th F-T cycle.

Two frequency packets are apparent in the series of graphs in Figure 10, describing the behaviour of CS150 test specimens during the last 25 F-T cycles, the more significant one at 50–120 kHz and the less significant one at 120–240 kHz. The maximum amplitude value occurs as late as 0.7 ms from the beginning of the signal. From that point in time, attenuation at the most significant frequency (75 kHz) can be determined—attenuation of spectral density is 6 dB/(Hz∙ms).

The typical AE signals of both test specimen sizes during the entire 100 F-T cycles (Figure 5 to Figure 10) confirm the conclusions inferred from the graphs in Figure 3 and Figure 4. That is, the amplitude decreases with the increasing number of F-T cycles and the dominant frequency shifts towards lower values.

The time-frequency power spectra also indicate a gradual decrease of the spectral density attenuation values, for CS100 specimens it is gradually 62, 11 and 4 dB/(Hz∙ms), for CS150 specimens then 43, 6 and 6 dB/(Hz∙ms). This decrease in attenuation is probably caused by the progressing damage to the internal structure of the concrete, or to its cementitious matrix, where the compromised matrix bonds that have been oscillated cannot be easily and quickly attenuated. Since they are loose, they oscillate slightly longer than at the beginning of the frost resistance test. In addition, there is a decrease in the frequency and amplitude of the generated AE signals. All these phenomena then result in an accelerating decrease of the spectral density attenuation values.

## 4. Discussion

There have not been many papers published recently that deal with the continuous monitoring of the behaviour of concrete specimens during F-T cycles. The papers that describe continuous measurement using the AE method, and at the same time, concern cement-based materials deal only with standard AE parameters, see, e.g., [44,45,46]. Other studies deal with different materials, such as water-saturated ceramics ([47], this study utilizes equipment from the same manufacturer as in the manuscript submitted by us) or materials for asphalt roads, see [48]. However, the acoustic emission method is typically used to compare the behaviour of materials after F-T cycles during mechanical loading, which is described, for example, in [49,50,51,52]. No other known study showed a frequency analysis of individual AE hits that recorded directly during F-T cycles.

## 5. Conclusions

The abovementioned results from the typical test procedures indicate that no significant changes in the quality of the concrete were recorded after 100 F-T cycles. The dynamic modulus of elasticity exhibited a maximum decrease of less than 3 percentage points (for RDM(FL) of the P95 specimen set after 50 F-T cycles) and mere 1.6 percentage points after 100 F-T cycles (also for RDM(FL) of the P95 specimen set). There was no decrease in tensile splitting strength due to freezing and thawing. The only standard parameter that exhibited some decrease was flexural strength. In this case, the decrease to 85.3% of the original value can be partly contributed to one test specimen of three test, which achieved lower strength. This confirms, inter alia, the problem with evaluation of concrete on the basis of a comparison of results obtained on different test specimens.

Based on both RS and RDM, the test specimens therefore appear to be almost intact by frost. Even the commonly presented cumulative number of AE counts does not indicate any significant difference between smaller (CS100) and larger (CS150) test specimens. However, a detailed analysis of the recorded AE signals that were generated directly during the F-T cycles in the test specimens proves the situation is different:There is a decrease in the average amplitude during freezing cycles in the case of both specimen sizes;There is an apparent difference between larger and smaller specimens in the height of the amplitude after between 75 and 100 F-T cycles;There is an evident shift in the values of the dominant frequency to lower values in the case of both specimen sizes;There is a noticeable difference between larger and smaller specimens in the shift of the dominant frequency from the 75th to the 100th F-T cycle;There is a decrease in the spectral density attenuation values during F-T cycles, with a higher decrease observed in the case of smaller CS100 test specimens.

It has been demonstrated that while conventional test procedures do not reveal significant changes in the internal structure of the tested concrete even after 100 F-T cycles, changes still occur in the material and can subsequently lead to irreversible damage of the material. It is, thus, appropriate to employ not only traditional test procedures, but also less traditional ones, such as the AE method, which can safely detect the formation of changes (damage) in the structure of concrete already at the beginning.

The innovative contribution of the manuscript lies in the application of the methods of frequency analysis of recorded AE hits during F-T cycles. It was discovered that the typical AE parameters (such as the cumulative number of counts) and commonly used construction testing methods are not sensitive enough to determine the degree of damage of concrete specimens during F-T cycles. In particular, in the case of air-entrained concrete, which exhibits very low degree of damage, is it necessary to employ modern tools for the analysis of AE hit signals. The submitted manuscript provides one of the first insights into the issue of structural health monitoring. It is possible that some testing procedures and methods will have to be revised in the future since the developments in building materials are still advancing and some current diagnostic tools will no longer be sufficient for the detection of developing damage. This manuscript could be one of the pioneers in this context.

## Figures and Tables

**Figure 1 materials-14-01230-f001:**
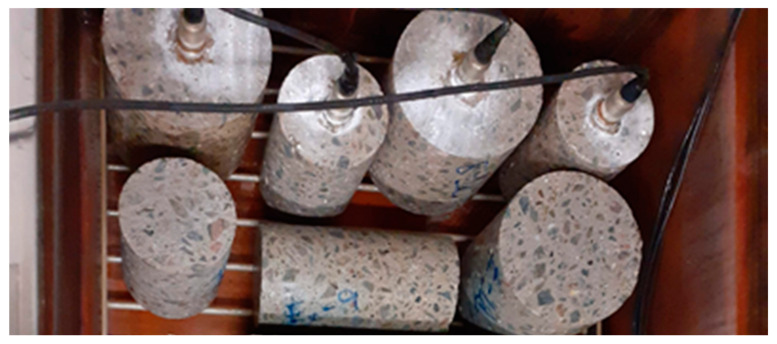
Arrangement of the test specimens with AE sensors in the KD 20 equipment during F-T cycles.

**Figure 2 materials-14-01230-f002:**
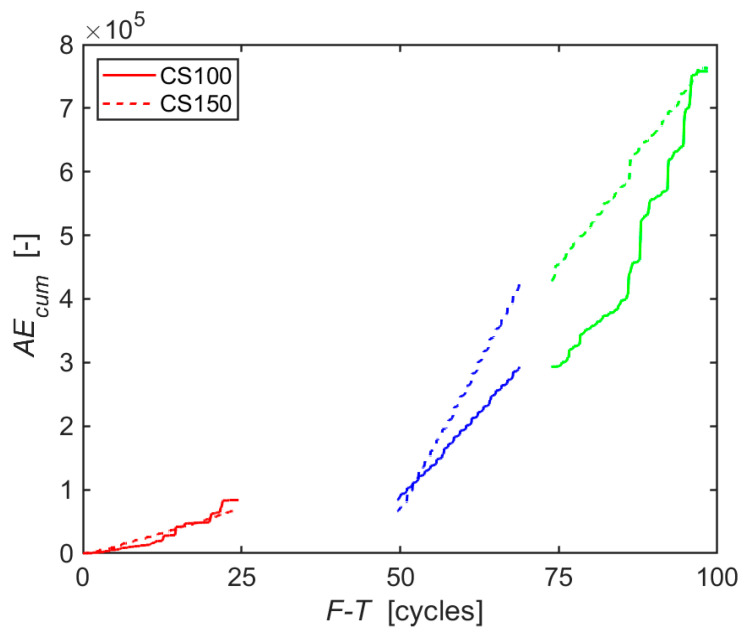
Cumulative number of AE counts during F-T cycles.

**Figure 3 materials-14-01230-f003:**
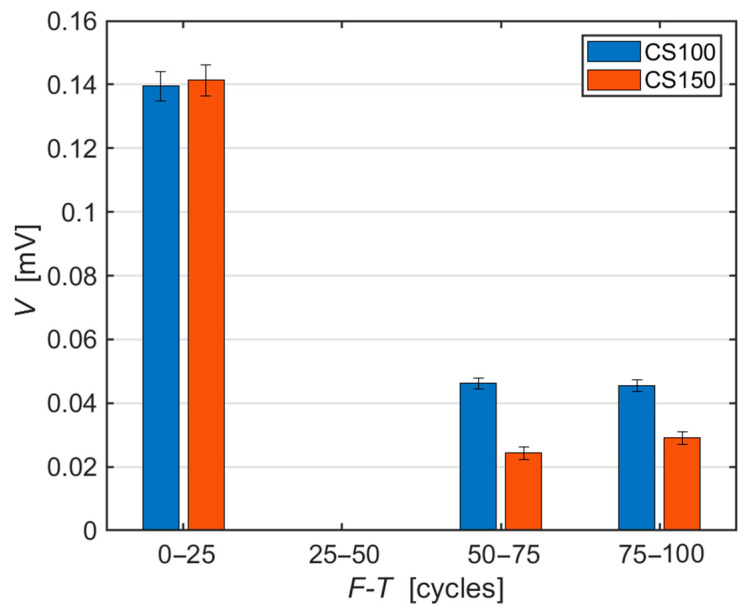
Average amplitude height of AE hits during F-T cycles, error bars represent sample standard deviations.

**Figure 4 materials-14-01230-f004:**
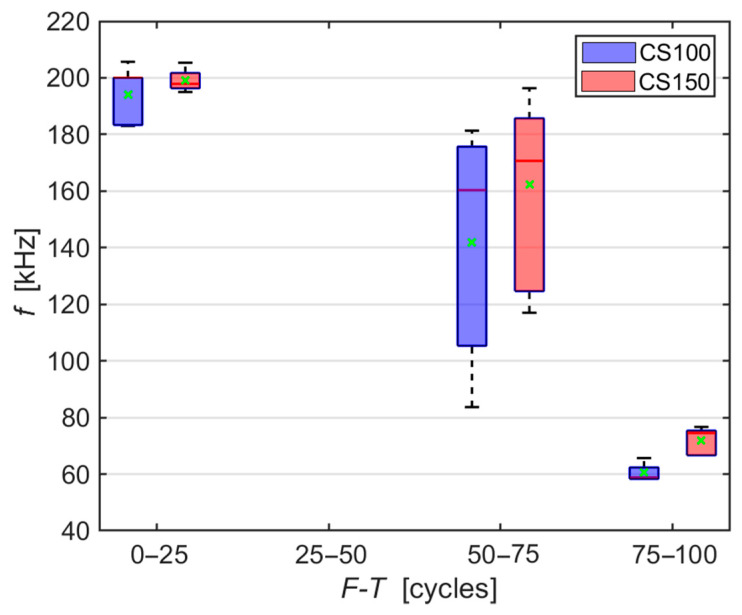
Box plot of development of dominant frequency values of AE hits during F-T cycles.

**Figure 5 materials-14-01230-f005:**
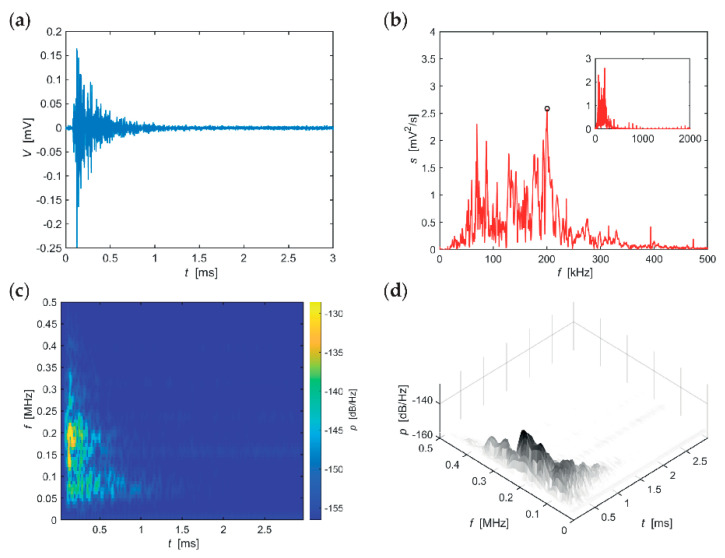
Visualisation of the recorded AE signal during 0–25 F-T cycles from specimens CS100: (**a**) time course of the amplitude, (**b**) frequency spectrum, (**c**) spectrogram STFT-2D image, (**d**) spectrogram STFT-3D image.

**Figure 6 materials-14-01230-f006:**
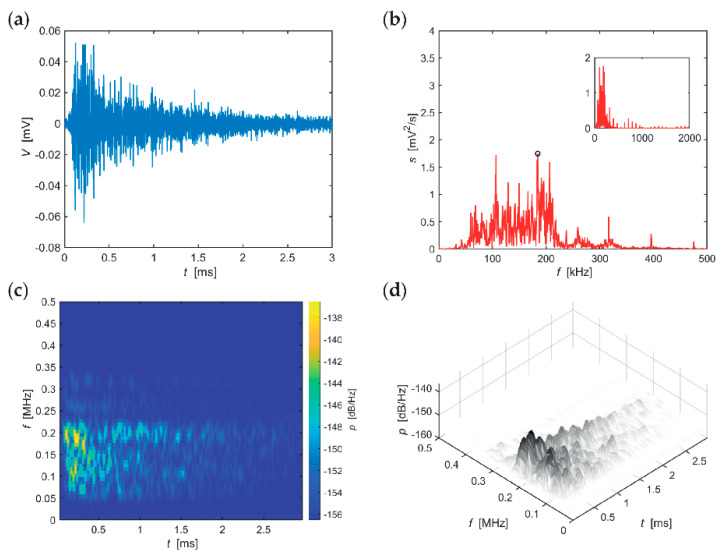
Visualisation of the recorded AE signal during 50–75 F-T cycles from specimens CS100: (**a**) time course of the amplitude, (**b**) frequency spectrum, (**c**) spectrogram STFT-2D image, (**d**) spectrogram STFT-3D image.

**Figure 7 materials-14-01230-f007:**
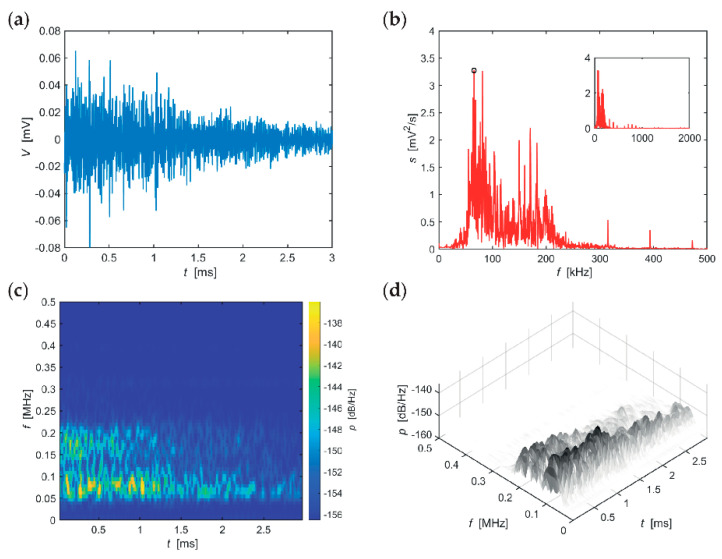
Visualisation of the recorded AE signal during 75-10 F-T cycles from specimens CS100: (**a**) time course of the amplitude, (**b**) frequency spectrum, (**c**) spectrogram STFT-2D image, (**d**) spectrogram STFT-3D image.

**Figure 8 materials-14-01230-f008:**
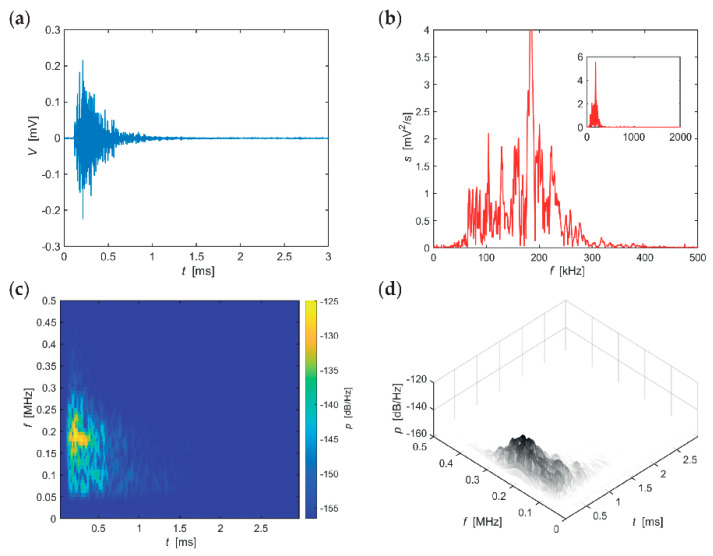
Visualisation of the recorded AE signal during 0–25 F-T cycles from specimens CS150: (**a**) time course of the amplitude, (**b**) frequency spectrum, (**c**) spectrogram STFT-2D image, (**d**) spectrogram STFT-3D image.

**Figure 9 materials-14-01230-f009:**
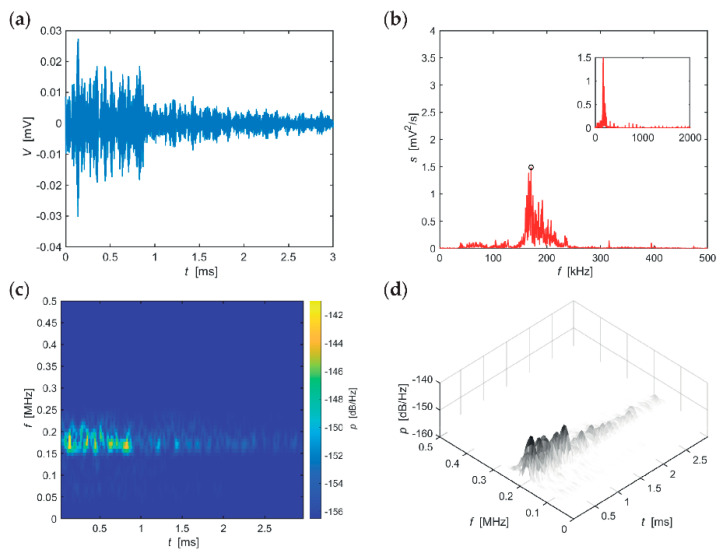
Visualisation of the recorded AE signal during 50–75 F-T cycles from specimens CS150: (**a**) time course of the amplitude, (**b**) frequency spectrum, (**c**) spectrogram STFT-2D image, (**d**) spectrogram STFT-3D image.

**Figure 10 materials-14-01230-f010:**
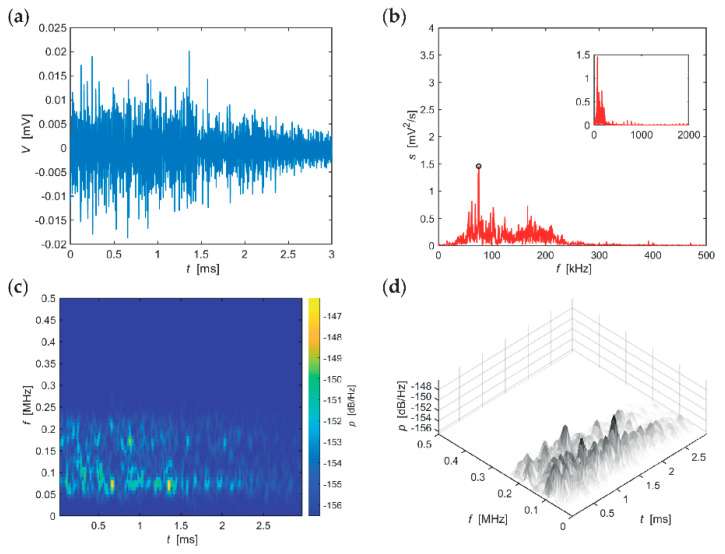
Visualisation of the recorded AE signal during 75–100 F-T cycles from specimens CS150: (**a**) time course of the amplitude, (**b**) frequency spectrum, (**c**) spectrogram STFT-2D image, (**d**) spectrogram STFT-3D image.

**Table 1 materials-14-01230-t001:** Theoretical composition of the concrete.

Component	kg/m^3^ of Fresh Concrete
Cement CEM I 42.5 R (HeidelbergCement Group, Mokrá, Czech Republic)	390
Sand 0–4 mm (HeidelbergCement Group, Tovačov, Czech Republic)	810
Natural crushed aggregate 4–8 mm (HeidelbergCement Group, Luleč, Czech Republic)	160
Natural crushed aggregate 8–16 mm (HeidelbergCement Group, Olbramovice, Czech Republic)	760
Water	185
Superplasticising admixture (Sika CZ, Brno, Czech Republic)	1.0
Air-entraining admixture (Sika CZ, Brno, Czech Republic)	0.6
Workability enhancing admixture (Sika CZ, Brno, Czech Republic)	1.6

**Table 2 materials-14-01230-t002:** Average tensile splitting f_ct_ and flexural f_cf_ strengths of the test specimens after 0, 50 and 100 F-T cycles.

Strength-Specimen Set	Number of F-T Cycles		
	0 (REF)	50	100
f_ct_-CS100	3.55	3.45	3.60
f_ct_-CS150	3.20	3.25	3.25
f_cf_-P95	5.1	4.6	4.3

**Table 3 materials-14-01230-t003:** Relative development of the monitored strengths and moduli of elasticity of concrete in relation to the number of F-T cycles in percent.

RS/RDM-Specimen Set	Number of F-T Cycles			
	25	50	75	100
RS(TS)-CS100	-	97.2	-	101.4
RS(TS)-CS150	-	101.6	-	101.6
RS(F)-P95	-	91.7	-	85.3
RDM(U)-P95	99.5	100.3	100.3	101.8
RDM(U)-C200	99.3	100.9	101.8	102.2
RDM(FL)-P95	98.1	97.1	97.8	98.4
RDM(FL)-C200	98.2	98.8	99.3	99.9

**Table 4 materials-14-01230-t004:** Slope of the line interpolated with cumulative AE counts during F-T cycles (the number after the slash represents the coefficient of determination R^2^).

Set of Specimens	F-T Cycles			
	0–25	25–50	50–75	75–100
CS100	540/0.922	-	1622/0.998	3086/0.947
CS150	433/0.997	-	2768/0.999	2070/0.996

## Data Availability

The data presented in this study are available on request from the corresponding author. The data are not publicly available due to ongoing research.

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
