# Peer review of "Analysis of Acoustic Emission Signals Recorded during Freeze-Thaw Cycling of Concrete"

_materials, 2021, doi:10.3390/ma14051230_

Round 1

Reviewer 1 Report

Dear  Authors,

In my opinion, the manuscript titled Analysis of acoustic emission signals recorded during freeze-thaw cycling of concrete presents original and valuable research and could be interesting for readers of the Sensors MDPI Journal, but due to the listed below drawbacks my recommendation is "Reconsider after major revision".

Please refer to the comments/remarks listed below:

  • "One FT cycle consists of freezing in air at -15 ° C to -20 ° C and thawing in a water bath at + 20 ° C. (...) The total time of one FT cycle is therefore 6 hours + approximately 15 "- what was the reason that such a scenario was chosen, please explain it.
  • Please comment on the impact of the durability of the AE sensor attachment to the sample surface on the AE signals measurements. Has the effect of repeated freezing-thawing been experimentally verified? 
  • In my opinion the commentary on the results, the found dependencies and conclusions were formulated for a too-small statistical sample. It is difficult to say what is the repeatability of the conducted experiments and their results. I think that, it is necessary to perform identical tests on at least 3 samples subjected to the same F-T cycles. Please give your opinion on this.

Author Response

Hello,

Thanks for the review. Please find our reactions below:

  1. We used the procedure defined by the Czech standard ČSN 73 1322 "Determination of frost resistance of concrete" and we did so for several reasons. The selected procedure is less time demanding than, for example, the procedure according to CEN/TR 15177, the temperature range is larger than, for example, in the procedure according to ASTM C666/C666M - 15 (which theoretically accelerates the concrete degradation process) and the freezing and thawing process can be very easily repeated since the times of the individual cycle parts are strictly defined. The negative temperature we selected was always -18°C.
  2. The AE sensors, including fine-tuning of their mounting, were thoroughly tested in 2013, well before the first freezing experiments were conducted, see doi: 10.1134/S1061830914020065. The first experiments also employed mechanical mounting, which we subsequently abandoned as it was not necessary. The AE sensors are tested and inspected every year and their characteristics remain unchanged.
  3. At least 3 test specimens were used for all destructive tests (even more in the case of non‑destructive tests), see lines 93-109. Only continuous measurement of acoustic emission involved two specimens for each specimen size due to a limited number of AE sensors. Despite that, the results from both sensors on one specimen size did not differ statistically significantly.

    Kind Regards

    team of authors

Reviewer 2 Report

The reviewer is impressed with the amount of work that went into this research. That alone gives one hope that a paper worthy of the archival standing of a major academic journal such as this is possible. The paper cannot be accepted in the present form as it needs further improvements.

I. Impressions, impact, add to the knowledge

The article presents an interesting topic with research towards analysis of acoustic emission signals recorded during freeze thaw cycling of concrete, however, some points need to be clarified and summarized.

-Abstract: The text must be carefully revised. Some sentences contain mistakes (in the abstract: very general statements), whereas some sentences must be reworded as the English are “meaningless.” I strongly recommend that the authors retain the services of a professional editor. Many reputable companies offer these services.

- In a research paper, it is expected that the introduction section briefly explains the starting background and, even more important, the originality (novelty) and relevancy of the study is well established. Once this is done, the hypothesis and objectives of the study need to be addressed, as well as a brief justification of the conducted methodology.

- The introduction part does not have a flow or direction. It has too many different medical terminologies thrown randomly. Proper references need to be used rather than using others. Language can be improved. The sentences are half-constructed or incomplete so that the readers are expected to fend for themselves to understand their meaning.

- Author must be enriching the references with the latest developments in the field. Some of the recent references can be added. The authors have not paid attention to previous research papers and concerns.

Acoustic emission-based inspection framework for non-destructive and destructive test methods are not described with appropriate bibliography. It can be difficult to read as numerous steps are used to establish the framework and some paragraphs are quite unclear to me.

II. Suggestions

Section 3 need to be rewritten. Just illustrating results (Fig 5 to 10), looks like a thesis report rather than the research article.

Table 4 – I don’t understand what the authors want to convey.

Eqs. (1) - (5) are well-known to the research community. Give proper citations.

Page 5: Line 199: You have used Tab 2 and Table 3. Try to use consistently as per the guidelines.

- The innovation contribution of this article is not clearly stated. The research contributions should be highlighted in the revised manuscript.

- Few statements have been repeated multiple times. Almost every figure needs to be updated. It is of very poor quality. What is more, the paper is difficult to understand at points.

Discussion Section: Introduce a new section "Discussion", with more current references, which compare the results obtained by the authors with other studies carried out by other researchers.

Conclusions Section: Improve the conclusions section, it is very general and does not clearly explain the main objectives achieved in this research.

The list could go on, but the bottom line is that the authors need to rewrite the paper or even reconsider the research content before it could be considered for publication in this journal. Since this work fails to provide convincing innovation and lacks in-depth viewpoints, I would recommend ‘Major revision’ for this paper.

Author Response

Dear reviewer,

We appreciate your review, but we are, unfortunately, not able to meet all your requirements since the manuscript was created over a period of several weeks and it is not possible to completely rewrite it in 10 days. Due to certain administrative complications, we already have 8 reviews and rest assured you are the only one who has issues with the comprehensibility of the text, which we do regret. We appreciate your insights and thank you for your comments.

Table 4 – I don’t understand what the authors want to convey.

The table offers a better understanding of ​​the slope of the lines in the graph in Fig. 2, for some colleagues, the table is clearer than a graphical presentation. The data in Tab. 2 numerically represent the changes in the slope of the individual curves.

Eqs. (1) - (5) are well-known to the research community. Give proper citations.

The manuscript now includes equation references.

Page 5: Line 199: You have used Tab 2 and Table 3. Try to use consistently as per the guidelines.

This issue has been unified.

Kind Regards

team of authors

Reviewer 3 Report

The work is interesting but it looks to me that the background study was not investigated well. There are many studies in this area worldwide. Most, if not all, of the references used are local to authors. Please use some international references. One example, effects of large freeze-thaw cycles on stiffness and tensile strength of asphalt concrete, ASCE Journal of Cold Regions Engineering. There are some editorial issues but those will be solved automatically during the publication process. The conclusion section  can be rewritten to concisely give the conclusion. In current state, this section became a discussion section. 

Author Response

Hello,

Thank you for the review. As we have now stated in the incorporated Chapter 4. Discussion, we have not found many papers that would deal with monitoring of concrete test specimens using the AE method directly during F-T cycles. If you are aware of any specific ones, please email them to us in the form of a link to: libor.topolar@vutbr.cz. Thank you in advance.

Kind Regards

team of authors

Round 2

Reviewer 1 Report

Hello,

thank you for your responses. Most of my doubts have been dispelled. The presented explanations justify the assumptions made in the research.

Please consider adding to the article text condensed comments on my previous remarks. In my opinion this will make the article clearer to readers. 

Author Response

Hello,

Thank you for your comments. We added a new text based on your suggestions:
lines 125 - 125
lines 127 - 130
line 134
lines 162 - 166.

Kind Regards
team of authors

Reviewer 2 Report

The paper has been improved. It can be accepted in the present form.

Author Response

Hello,
Thank you for the final comment.

Kind Regards
team of authors

This manuscript is a resubmission of an earlier submission. The following is a list of the peer review reports and author responses from that submission.